# Fecal shedding of *Salmonella* spp., *Clostridium perfringens*, and *Clostridioides difficile* in dogs fed raw meat-based diets in Brazil and their owners' motivation

**Flavia Mello Viegas**, **Carolina Pantuzza Ramos**, **Rafael Gariglio Clark Xavier**, **Emily Oliveira Lopes**, **Carlos Augusto Oliveira Júnior**, **Renata Marques Bagno**, **Amanda Nadia Diniz**, **Francisco Carlos Faria Lobato**, **Rodrigo Otávio Silveira Silva** \*

Department of Preventive Veterinary Medicine Federal University of Minas Gerais (UFMG), Belo Horizonte, Minas Gerais, Brazil

\* rodrigo.otaviosilva@gmail.com

**Data Availability Statement:** All relevant data are within the paper and its Supporting Information files.

## Abstract

The present study aimed to explore the motivations of Brazilian dog owners and their knowledge about the risks related to raw meat-based diets (RMBD) as well as to evaluate important enteropathogens such as *Salmonella* spp., *C. perfringens*, and *C. difficile*, in feces of dogs fed different diets. The majority of the pet owners (69.3%) reported to have chosen this diet for their dogs, considering it to be more "natural". A large number of owners declared that RMBD do not pose health risks for their animals (87.9%) or humans (98.8%), even though almost one third of the respondents (34.8%) declared having at least one individual at high risk of infection in contact with RMBD-fed dogs. Stool samples from 46 RMBD-fed dogs and 192 dogs fed commercial dry feed were collected. The present study revealed that dogs fed raw meat diets were almost 30 times more likely to be positive for *Salmonella* spp. than dogs on a conventional diet. Some of the serovars detected were commonly associated with human salmonellosis, such as *S*. Typhimurium and *S*. Saintpaul, and were multidrug resistant. RMBD-fed dogs were more likely to be positive for *C. perfringens* type A ($p = 0.008$) and one *C. perfringens* type F was isolated from these animals. Two toxigenic strains (4.3%) of *C. difficile* were isolated only from raw meat-fed dogs, all of which were under antibiotic therapy. These toxigenic *C. difficile* isolates were classified as RT106/ST54 and RT600/ST149, previously associated with infection in dogs and humans. The present work revealed that the owners have a tendency to ignore or are unaware of the risks associated with raw meat diets for dogs. Also, the higher fecal shedding of important enteropathogens in dogs fed RMBD suggests that this diet poses a risk for the animals and the people in contact with them.

## Introduction

The number of dogs and cats in households have increased worldwide. In Brazil, the latest census of companion animals has shown a population of approximately 52.2 million dogs, being

**Funding:** This worked was supported by the Minas Gerais Research Support Foundation (Fundação de Amparo à Pesquisa do Estado de Minas Gerais - FAPEMIG), the Brazilian Federal Agency for the Support and Evaluation of Graduate Education (Coordenação de Aperfeiçoamento de Pessoal de Ensino Superior - CAPES) grant Prêmio CAPES 2015 to ROSS, the National Council for Scientific and Technological Development (Conselho Nacional de Desenvolvimento Científico e Tecnológico - CNPq), and Pró-Reitoria de Pesquisa, Universidade Federal de Minas Gerais to ROSS. The funders had no role in study design, data collection and analysis, decision to publish, or preparation of the manuscript.

**Competing interests:** The authors have declared that no competing interests exist.

the country with the 2nd largest number of these domesticated animals [1]. The relationship between the pet owner and their dogs has also changed, with an increasing view of dogs as family members [2].

In recent years, a large number of owners have been feeding their dogs and cats raw meat-based diets (RMBD) instead of regular commercial dry feed [3,4]. Studies have shown that dogs fed RMBD have increased fecal shedding of some zoonotic enteropathogens, including *Salmonella* spp. and *Campylobacter jejuni* [5–7]. Moreover, a recent study showed the presence of other zoonotic pathogens in commercial raw meat diets for dogs, including diarrheagenic *Escherichia coli*, and *Listeria monocytogenes*, suggesting a risk for pet health as well as for the owner, who manipulates these products [8].

In light of these reports, veterinary and public health organizations, including Centers for Disease Control and Prevention (CDC), World Small Animal Veterinary Association (WSAVA) and US Food & Drug Administration (FDA), have published statements discouraging the inclusion of raw meat in the diets of dogs and cats [9–11]. Despite these reports, the popularity of RMBD seems to be increasing worldwide, while little is known regarding the motivation and characteristics of the owners who feed their dogs this diet. Moreover, there are few studies study focusing on other relevant human and dog enteropathogens, including *Clostridium perfringens* and the emergent zoonotic pathogen *Clostridioides* (previously *Clostridium*) *difficile*. Such information will help understand the trends associated with this increased popularity of RMBD and risks associated with this practice [4].

Therefore, the present study aimed to elucidate the motivations of Brazilians who fed their dogs RMBD and to clarify if they acknowledged the risks associated with this practice. Additionally, the presence and antimicrobial resistance profile of *Salmonella* spp., *C. perfringens*, and *C. difficile* in dogs fed RMBD and dogs fed commercial dry feed were evaluated.

## Material and methods

### Web-based survey

In order to characterize Brazilian pet owners and their motivations for feeding RMBD, an anonymous web-based survey was developed in Google Forms (docs.google.com/forms) opened from November 2017 to March 2018. Similar to previous reports [4,12,13], participants were recruited through posts on a social media platform (Facebook) in groups related to dog feeding and dog breeding and also through personal requests to students and staff at the Federal University of Minas Gerais. Dog owners ≥18 years of age were eligible to participate after agreeing with the online participant consent, as previously approved Research Ethics Committee of the Faculty of Medicine of Federal University of Minas Gerais (CAAE 89680318.0000.5149). The questionnaire containing 12 closed questions also based on Morgan et al. [4] and Morelli et al. [12]. Participants were interrogated on the following topics: (I) diet provided for their animals; (II) main reason for adopting RMBD; (III) recognition of risks associated with RMBD for the animal as well as for the owners themselves; (IV) contact of the animals with individuals at high risk for salmonellosis (people younger than 5 years or older than 65 years, pregnant or immunocompromised, according to Gerba et al. [14]; (V) occurrence of diarrhea in the last 6 months. A copy of the web-based survey is available as a supporting information (S1 File).

### Stool samples

A non-probabilistic sampling of dogs fed RMBD or commercial dry feed in Minas Gerais state, southeastern region of Brazil, were made from December 2017 to July 2018. Owners that claimed to provide raw meat diets for their animals were invited to participate in the study.

Part of these owners were invited after the web-survey and part were indicated by other owners or veterinarians. A total of 46 RMBD-fed dogs were included. Each stool sample were collected by the owner and submitted to the laboratory analysis no later than 24 hours after defecation. The owners were invited to answer a short questionnaire regarding their animals' health. Four owners (4/46 = 8.6%) reported that their dogs were undergoing antibiotic therapy during the collection of the samples and five (5/46 = 10.8%) were diarrheic at the moment of the collection. Similar to previous studies, 192 stool samples from dogs fed commercial dry feed were obtained in city squares in Belo Horizonte (Minas Gerais, Brazil), with prior permission of the owner, and only fecal material that did not have contact with the environment was collected [15,16]. None of these animals were diarrheic and, according to the owners, only one (0.5%) had undergone antibiotic therapy in the last month, prior to sample collection. All stool samples were immediately stored in a cooler with ice packs and transported to the Bacteriology and Research Laboratory, School of Veterinary of Universidade Federal de Minas Gerais (UFMG). This study was approved by Ethical Committee on Animal Use (CEUA–UFMG) under the protocol 51/2015.

## Isolation and serotyping of *Salmonella* spp

For the isolation of *Salmonella* spp., each stool sample was inoculated into Tetrathionate Broth (Oxoid, USA), followed by plating on Hektoen agar (Oxoid, USA) [17]. Sulfite-reducing colonies were subjected to a previously described PCR analysis [18] and strains confirmed as *Salmonella* spp. were differentiated into species, subspecies [19] and finally, serotypes, by antigenic characterization, based on the White-Kauffmann-Le Minor scheme. The antigenic characterization was performed by slide agglutination with somatic (O), flagellar (H), and occasionally, capsular (Vi), poly and monovalent antisera and prepared at the LABENT, the Oswaldo Cruz Institute (FIOCRUZ), Rio de Janeiro, Brazil. The identification of the specific serovar was performed and represented according to the criteria reported by Grimont and Weill [20].

## Antimicrobial susceptibility of *Salmonella* spp. isolates

To characterize antimicrobial susceptibility of *Salmonella* spp., the disc diffusion method was applied as recommended by the Clinical and Laboratory Standards Institute (CLSI) [21]. Using Mueller Hinton agar (Difco Laboratories, USA), the inhibition zone sizes were interpreted as per the VET01-S2 guidelines [21]. Representing seven different classes of antimicrobial agents, the antibiotic disks used were trimethoprim/sulfamethoxazole (25 μg), ceftriaxone (30 μg), cephalothin (30 μg), streptomycin (10 μg), enrofloxacin (5 μg), gentamicin (10 μg), amoxicillin/clavulanic acid (30 μg), metronidazole (50 μg) and oxytetracycline (30 μg) (DME, Brazil).

## Isolation and genotyping of *Clostridium perfringens*

For the isolation of *C. perfringens*, stool samples were diluted 1:10 in 0.9% saline solution, and aliquots of 10 μL of each dilution were plated onto Shahadi Ferguson Perfringens agar (SFP, Difco Laboratories, USA) and anaerobically incubated at 37˚C for 24 hours [22]. At least three rounded sulfite-reducing colonies from each dilution were subjected to a previously described PCR protocol [23] for the detection of genes encoding the following *C. perfringens* toxins: alpha, beta, epsilon, iota, enterotoxin, and beta-2 toxin [22]; NetB [24]; NetE, NetF, and NetG [25].

### *Clostridioides difficile* isolation and A/B toxin detection

To isolate *C. difficile*, equal volumes of stool samples and 96% ethanol (v/v) were mixed and incubated for 30 minutes at room temperature. Thereafter, 20 μL aliquots were inoculated on plates containing cycloserine-cefoxitin-fructose agar, supplemented with 7% horse blood and 0.1% sodium taurocholate (Sigma-Aldrich Co., USA) [22]. Following anaerobic incubation at 37˚C for 72 hours, all colonies with suggestive morphology (flat, irregular, and with ground-glass appearance) were subjected to a previously described multiplex-PCR for a housekeeping gene (*tpi*), the toxin A gene (*tcdA*), the toxin B gene (*tcdB*), and a binary toxin gene (*cdtB*) [26]. *C. difficile* isolates were PCR ribotyped as previously described [27]. In addition, toxigenic *C. difficile* strains were submitted to multilocus sequence typing (MLST), as previously described [28]. The amplicon sequences, so generated, were compared with the MLST data-base (https://pubmlst.org/cdifficile/) to identify the allelic profiles and the corresponding sequence type (ST). Stool samples positive for toxigenic *C. difficile* isolation were subjected to toxin A/B detection by an enzyme-linked immunosorbent assay (EIA) kit (*C. difficile* Tox A/B II, Techlab Inc., USA).

### *Clostridioides difficile* antimicrobial susceptibility

The minimal inhibitory concentrations (MIC) of metronidazole, vancomycin, clindamycin and ciprofloxacin were determined by gradient test with the M.I.C. Evaluator™ strips (M.I.C. E.™, Oxoid, USA). Briefly, a suspension of *C. difficile* was prepared in sterile 0.9% saline, from a pure culture after 24 hours' growth in Brucella agar, using McFarland standard 1 as the reference. The test was performed on Brucella agar (Oxoid, USA) with 5% lysed blood, supplemented with hemin (Difco Laboratories, USA) and vitamin K (Sigma-Aldrich Co, USA). Plates were incubated at 37˚C in an anaerobic atmosphere, and the MIC were measured after 48 hours of incubation. The MIC values were interpreted according to clinical breakpoints from the CLSI and EUCAST guidelines [29–31].

### Statistical analysis

To measure the association among the categorical variables (questionnaire answers, diet type, occurrence of diarrhea and antimicrobial use) and the isolated strains, a univariate analysis using the Chi-square test or Fisher's exact test was performed, employing an alpha error of 5%. All analyses were performed using Stata 12® software (StataCorp, USA).

## Results and discussion

### Web-based survey

Among the 412 participants, 246 (59.7%) reported to have fed their animals only with commercial dry feed, while 166 (40.3%) fed their pets RMBD (S1 File). Interestingly, 107 (64.5%) of the owners who fed their pets RMBD had adopted this diet less than one year ago, while 37% of those who reported to have fed only commercial dry feed declared their intention of changing their pets' diet to RMBD in the future. These results suggest that, despite the known possible dietary and sanitary risks associated with this diet [6–8,11,32], the adoption of the RMBD is on an incline, similar to that recently reported in a study in the US [4].

Interestingly, the main reason for majority of the pet owners (69.3%) to adopt the RMBD for their pets, was because it respected their "animal nature" (Fig 1). This finding was similar to those of previous surveys conducted in the US and Italy [4,12]. It is known that domestication has induced certain changes in the canine gut microbiota, resulting in an increased ability to metabolize fibers and carbohydrates [6,12]. Despite the absence of studies suggesting

a) Main reasons associated with adoption of RMBD for dogs in Brazil (n = 166)

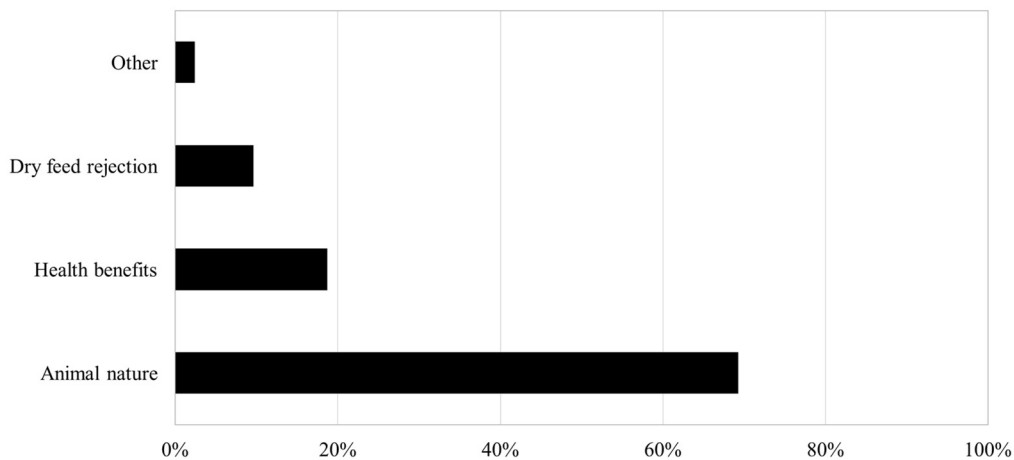

b) Individuals at increased risk of *Salmonella* spp. infection and in contact with RMBD-fed dogs (n = 166)

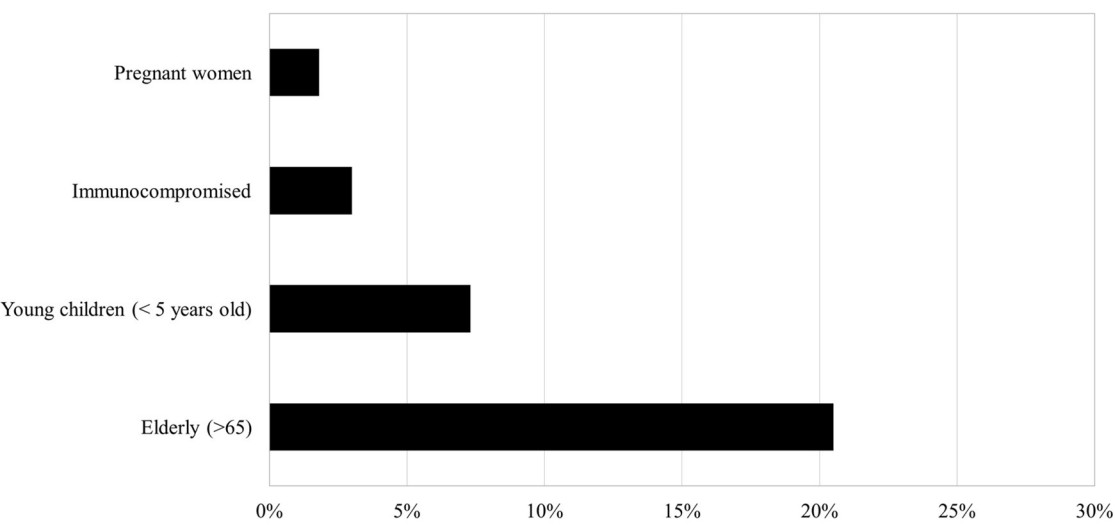

c) Perception of risks for humans and animals associated with RMBD for dogs (n = 166)

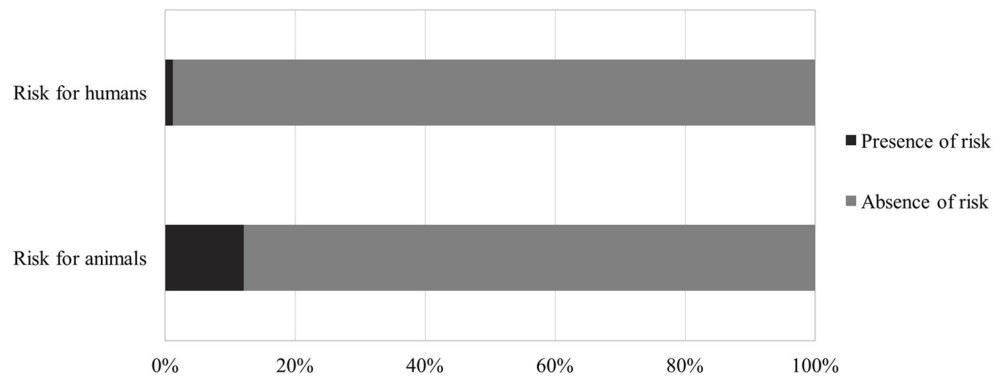

**Fig 1.** Main reasons associated with adoption of RMBD for dogs in Brazil (n = 166) (a), individuals in increased risk of *Salmonella* spp. infection in contact with RMBD-fed dogs (n = 55) (b), perception of risks for humans and animals associated with RMBD for dogs (n = 166) (c).

benefits of the RMBD [6,11], 18.7% of the owners indicated that health benefits was the main reason for the adoption of this practice (Fig 1). In fact, previous studies have shown that proponents of RMBD claim that these diets are healthier and more natural for dogs and cats than conventional pet food [6,33,34].

The present study revealed that majority of the Brazilian owners believe that the RMBD does not represent a risk for their pets (87.9%) (Fig 1). Morelli et al [12] also reported that majority of the interviewed owners of RMBD-fed dogs in Italy did not recognize the health risks associated with this kind of diet, although the usual occurrence of diarrhea in their dogs, after beginning of RMBD, was observed. Curiously, almost one third of the owners of RMBD fed dogs in the present study (53/166 = 31.9%), reported the occurrence of diarrhea in their animals in the past six months. In fact, several studies have reported this syndrome in animals fed with raw meat diets, which is commonly associated with increased shedding of *Salmonella* spp. [6,7]. However, no significant difference was noted between the reports of diarrhea in dogs fed with RMBD and those fed with dry food (S1 File).

Regarding the risks that RMBD posed for the owners themselves, all but two owners (98.8%) believed that handling and feeding their dogs with raw meat does not represent a risk for their health (Fig 1). However, 39 (34.8%) participants reported having at least one person at risk of salmonellosis in the household, similar to that previously reported [12] (Fig 1). Therefore, even though more than one third of the dogs on RMBD are housed with high risk individuals, majority of their owners have a tendency to ignore or are unaware of danger that this diet poses for their pets, themselves, or other individuals sharing the same residence. In fact, it is known that dogs fed with these natural diets are at an increased risk of shedding pathogenic agents in their feces, exposing all individuals at risk, for infection [6,13]. The failure to acknowledge the human health risks associated with raw meat diets was recently reported by Alho et al. [35] in a survey to evaluate the awareness of the individuals who owned dogs and cats about zoonotic disease. This lack of knowledge is worrisome as it can reduce the compliance of basic hygiene necessary while handling raw meat, considering the potential risk of infection related to the handling of these raw diets [36,37]. Together, these results suggest an imperative need for strategies to warn the owners about the health risks associated with RMBD, for their pets as well as themselves. Persuading owners to alter feeding practice are known to be challenging [38]. In addition, recent studies indicated that majority of the owners who adopted RMBD had a low level of trust in veterinary advice with respect to nutrition, commonly relying on information sources other than veterinarians [4,6,12,39]. Therefore, further studies evaluating strategies to efficiently communicate with these owners seems to be an urgent necessity.

### *Salmonella* spp

*Salmonella* spp. is one of the most common zoonotic pathogens worldwide [40]. Although the majority cases in humans are foodborne [41], there is an increase concern regarding the role of pets as reservoirs for nontyphoidal *Salmonella* [42,43]. In the present study, 15.2% (7/46) dogs fed RMBD shed *Salmonella enterica*, including *S*. Typhimurium, *S*. Heidelberg, and *S*. Saintpaul, which are frequently associated with human salmonellosis [44,45]. Conversely, only one dog (1/192 = 0.5%) fed commercial dry feed was positive for *Salmonella* spp. (Table 1). Therefore, dogs fed RMBD were 30 times more likely to be positive for this zoonotic pathogen

**Table 1. Frequency of isolation of *Salmonella* spp., *Clostridium perfringens* and *Clostridioides difficile* from dogs fed commercial dry feed or "Raw meat-based diet" (n = 238).**

| Dogs diet | *Salmonella* spp. | *Clostridium perfringens* | | *Clostridioides difficile* | | |
|---|---|---|---|---|---|---|
| | | Type A | Type F | Isolation | Toxigenic (A⁺B⁺CDT⁻) | RT/ST |
| **Commercial dry feed (n = 192)** | 1 (0.5%) [b] | 92 (47.9%)[b] | 0 (0%) | 4 (2.1%) | 0 (0%) | - |
| **Raw meat-based (n = 46)** | 7 (15.2%) [a] | 32 (69.6%)[a] | 1 (2.2%)[1] | 2 (4.3%) | 2 (4.3%)[2] | 106/52 600/149 |
| **Total (n = 238)** | 8 (3.4%) | 124 (52.1%) | 1 (0.4%) | 6 (2.5%) | 2 (0.8%) | - |

Different lower-case letters indicate a significant difference ($p < 0.01$)

[1]Stool sample negative for enterotoxin (CPE) in a commercial EIA test and negative for NetF-encoding gene by PCR

[2]One stool sample were positive for A/B toxins in a commercial EIA test.

when compared to dogs fed commercial dry feed (Table 1). These results are in accordance with previous studies that have shown that the fecal shedding of *Salmonella* spp. seems to increase in dogs fed RMBD, which might increase the risk of salmonellosis in their owners [7,13,32,34,45]. In addition to the human health risk, some studies have suggested that raw meat diets can enhance the occurrence of diarrhea and *Salmonella*-related infection in dogs [7,45–47], sometimes even leading to the animal´s death [6,48]. Curiously, two out of the seven (28.5%) *Salmonella*-positive dogs were diarrheic. Further studies are necessary to elucidate the role of RMBD and its association with the occurrence of diarrhea in dogs.

Majority of the *Salmonella* spp. isolates (7/8 = 87.5%) were resistant to at least one of the seven different classes of antimicrobials tested (Table 2). Additionally, three *Salmonella enterica* of serovars *S.* Typhimurium, *S.* Panama, and *S.* Saintpaul, from dogs fed RMBD, were multidrug resistant (resistant to three or more antimicrobial classes, according to Sweeney et al. [49]). It is interesting to note that this resistance pattern included antimicrobials that are largely used for human and veterinary medicine, such as oxytetracycline, streptomycin, and trimethoprim/sulfamethoxazole [44]. Additionally, one *Salmonella* spp. isolate from a raw meat-fed dog was resistant to enrofloxacin, a fluoroquinolone considered critically important in veterinary medicine for treatment of enteric diseases and septicemia [50].

Increased occurrence of antimicrobial-resistant *Salmonella* infections in humans is directly associated with the use of antibiotics in livestock animals [51]. Moreover, previous studies suggest that the higher antimicrobial resistance profile observed in *Salmonella* isolates from dogs fed RMBD is due to large exposure to resistant *Salmonella* spp. from the raw animal products

**Table 2. Serotypes of *Salmonella enterica* not susceptible for different antimicrobial agents isolated from dogs feed commercial dry feed diet (n = 1) and raw-meat based diet (n = 7).**

| Serotype | No. of isolates | Dog diet | Antimicrobial[1] |
|---|---|---|---|
| *S.* Typhimurium | 2 | Raw meat based | OXT, STX, MTZ, STR, ENR |
| | | Raw meat based | OXT, MTZ, STR |
| *S.* Saintpaul | 3 | Raw meat based | OXT, STX, MTZ |
| | | Raw meat based | Susceptible to all tested antimicrobials |
| | | Commercial dry feed | MTZ, STR |
| *S.* Schwarzengrund | 1 | Raw meat based | OXT, MTZ |
| *S.* Panama | 1 | Raw meat based | AMC, OXT, STX, MTZ, STR |
| *S.* Heidelberg | 1 | Raw meat based | STX, MTZ |

[1] Disc diffusion method according to Clinical and Laboratory Standards Institute (CLSI).

AMC, amoxicillin/clavulanic acid; ENR, enrofloxacin; MTZ, metronidazole; OXT, oxytetracycline; STR, streptomycin; STX, trimethoprim/sulfamethoxazole.

used in these diets [44,46]. In fact, studies have reported the isolation of resistant strains of *Salmonella* spp., including *S.* Heidelberg, *S.* Typhimurium, and *S.* Saintpaul, in raw food for dogs [46] and in animal carcasses, in several places, including Brazil [52]. Therefore, the present study highlights the increased shedding of antimicrobial resistant *Salmonella* spp. from RMBD-fed dogs, including multidrug resistant strains of serovars commonly associated with human salmonellosis.

## Clostridium perfringens

*C. perfringens* is a widespread gram-positive anaerobic bacillus, commonly found as part of the microbiota of animals and humans. This bacterium is classified into seven types (A to G) according to the production of six toxins: alpha, beta, epsilon, iota, and, more recently, enterotoxin and "necrotic enteritis like-B toxin" (NetB) [53]. In addition to these toxins, it can produce several additional virulence factors, including NetF, a recently described pore-forming toxin associated with acute enteritis in dogs and foals [15,25]. Despite strong evidence that NetF is the virulence factor associated with *C. perfringens* infection in dogs, the predisposing factors associated with the development of this disease is still largely unknown [54].

In the present study, dogs fed with RMBD were more likely to be positive for *C. perfringens* ($p$ = 0.008) (Table 1) than those receiving commercial dry feed. This result corroborates previous reports of metagenomic studies showing that dogs fed with natural diet were more likely to be positive for *C. perfringens* than those fed with commercial dry diet [55,56]. These results suggest that raw meat-based diet alters the *C. perfringens* population in the gut. Another possibility is the high contamination of the diet with *C. perfringens*; previous studies have reported a high isolation frequency of *C. perfringens* from raw pet food samples, which might contribute to the greater shedding of this agent from RMBD-fed dogs [57,58].

In order to better understand the possible differences in *C. perfringens* isolates from dogs fed dry feed and RMBD, all strains were genotyped and the presence of some important additional virulence factors were evaluated. Almost all isolates (123/124 = 99.2%) were genotyped as type A (Table 1), while one dog fed RMBD was positive for *C. perfringens* type F (simultaneously positive for alpha toxin and enterotoxin encoding genes). No isolate was found positive for the remaining virulence factors tested, including the NetF encoding gene (*netF*). Type F strains are associated with food poisoning and antibiotic associated diarrhea in humans [53], while type F strains positive for NetF-encoding gene are responsible for bloody diarrhea in dogs and foals [25]. Moreover, it is known that healthy dogs can carry *C. perfringens* type F [59]. In fact, this dog was apparently healthy, without diarrhea, during sample collection, and the enterotoxin itself (CPE) was not detected in the stool sample by a commercial EIA, suggesting that the dog was a carrier of this strain, rather than being infected by it.

## Clostridioides difficile

*C. difficile* is an anaerobic gram-positive bacterium that is considered an emerging pathogen, responsible for majority of the nosocomial diarrhea cases in humans [60]. In dogs, several studies report *C. difficile* as a cause of canine acute or chronic diarrhea [15,61]. In addition to its importance as a canine enteropathogen, recent studies have shown a high similarity between *C. difficile* isolates from humans and companion animals, suggesting a possible zoonotic transmission [62,63].

In the present work, no difference was observed in the *C. difficile* isolation rate from dogs fed with commercial dry feed or RMBD, suggesting that the latter did not lead to increased fecal shedding of *C. difficile*. Nevertheless, the two toxigenic *C. difficile* strains isolated in the present study were recovered from dogs fed RMBD (Table 1). Our results differ from that of a

study in kittens, which reported that *C. difficile* toxins were more likely to be detected in cats from the raw diet group [64]. Another recent study also showed a significant decrease in the *C. difficile* isolation rate in dogs and cats when fed with dry food [65]. Nevertheless, it is important to note that these two animals positive for toxigenic *C. difficile* were under antibiotic therapy when sampled, which is a known predisposing factor for *C. difficile* colonization [66]. In fact, we observed an association between antimicrobial therapy and *C. difficile* isolation in the present study ($p = 0.0001$); three out of the five dogs that were under antibiotic therapy during the sample collection were also positive for *C. difficile* isolation, two of which harbored toxigenic isolates.

Four of the five dogs undergoing antibiotic therapy during sample collection were being treated for causes unrelated to the gastrointestinal tract. The exception was a dog, fed RMBD, which received metronidazole and trimethoprim-sulfamethoxazole for pasty diarrhea. This animal was positive for toxigenic *C. difficile* and *Salmonella* Panama. Interestingly, its stool sample was also positive for A/B toxin in an EIA test, suggesting this animal was suffering from CDI, rather than only being a carrier of a toxigenic isolate. Some reports have shown *C. difficile* as a cause of infectious diarrhea in dogs, although its relevance as a canine pathogen is still uncertain [15,58]. Nonetheless, this agent is an important human pathogen and several studies suggest the role of animals, including dogs, as a potential source of community acquired CDI in humans, considering the high genetic similarity between dog and human isolates [63].

The toxigenic *C. difficile* isolates recovered in the present study were classified as *RT106/ST54* and *RT600/ST149* respectively, both of which are previously reported for causing infections in dogs and humans, in Brazil and other countries [15,63,67]. Furthermore, studies have shown that raw meat from several different production species are commonly contaminated with toxigenic *C. difficile* [66], indicating that diet could be a possible source of *C. difficile* spores for these animals. All the toxigenic *C. difficile* isolates in the present study were susceptible to metronidazole and vancomycin, the drugs of choice for treating CDI in humans, and to clindamycin and ciprofloxacin, antimicrobials that are known to increase the risk factors for CDI development [31,68]. Therefore, the present work confirmed that RMBD-fed dogs can harbor toxigenic *C. difficile* strains, including strain types previously described in CDI in humans, and reported in other studies with dogs fed with regular commercial diet [15,63].

## Conclusion

The main motivation reported by Brazilian owners, who adopted RMBD, is the belief that it is more "natural" for their animals. Moreover, the present work indicates that these owners are unaware or have a tendency to ignore the risks posed by this diet for their dogs as well as for humans. Microbiological analysis of the feces samples of the animals included in this study suggests that dogs fed RMBD are more likely to be positive for *C. perfringens* and *Salmonella* spp., including multiresistant strains and serovars previously described in humans as causative organisms for infections. The present work also revealed that these animals can harbor *C. difficile* isolates from ribotypes/sequence types previously reported for causing disease in humans. Taking all these results together, this study suggests that is imperative to better communicate with the owners about the risks that RMBD impose to the pets and themselves. In addition, the microbiological results also suggest the need for, at the very least, careful hygiene procedures for handling feces of pets fed RMBD.

## Supporting information

**S1 File. Questionnaire details (English).**
(DOCX)

**S2 File. Questionnaire details (Portuguese).**
(DOCX)

## Acknowledgments

The authors acknowledge the owners and their dogs participating in the study for making this work possible.

## Author Contributions

**Conceptualization:** Francisco Carlos Faria Lobato, Rodrigo Otávio Silveira Silva.

**Data curation:** Flavia Mello Viegas.

**Formal analysis:** Carolina Pantuzza Ramos, Rodrigo Otávio Silveira Silva.

**Funding acquisition:** Francisco Carlos Faria Lobato, Rodrigo Otávio Silveira Silva.

**Investigation:** Flavia Mello Viegas, Carolina Pantuzza Ramos, Rafael Gariglio Clark Xavier, Emily Oliveira Lopes, Carlos Augusto Oliveira Júnior, Amanda Nadia Diniz.

**Project administration:** Francisco Carlos Faria Lobato, Rodrigo Otávio Silveira Silva.

**Resources:** Flavia Mello Viegas, Renata Marques Bagno.

**Supervision:** Francisco Carlos Faria Lobato, Rodrigo Otávio Silveira Silva.

**Validation:** Flavia Mello Viegas, Carolina Pantuzza Ramos, Rafael Gariglio Clark Xavier, Emily Oliveira Lopes, Carlos Augusto Oliveira Júnior, Amanda Nadia Diniz.

**Visualization:** Flavia Mello Viegas, Carolina Pantuzza Ramos, Rodrigo Otávio Silveira Silva.

**Writing – original draft:** Flavia Mello Viegas, Carolina Pantuzza Ramos.

**Writing – review & editing:** Flavia Mello Viegas, Carolina Pantuzza Ramos, Rafael Gariglio Clark Xavier, Emily Oliveira Lopes, Carlos Augusto Oliveira Júnior, Renata Marques Bagno, Amanda Nadia Diniz, Francisco Carlos Faria Lobato, Rodrigo Otávio Silveira Silva.

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
