## [Decision Letter · Decision Letter 0]

12 Mar 2020

PONE-D-20-01523

Fecal shedding of Salmonella spp., Clostridium perfringens, and Clostridioides difficile in dogs fed raw meat-based diets in Brazil and their owners’ motivation

PLOS ONE

Dear Dr. Silva,

Thank you for submitting your manuscript to PLOS ONE. After careful consideration, we feel that it has merit but does not fully meet PLOS ONE’s publication criteria as it currently stands. Therefore, we invite you to submit a revised version of the manuscript that addresses the points raised during the review process.

Your work has been well assessed by reviewers, however, some minor revisions are required before its final acceptance.

We would appreciate receiving your revised manuscript by Apr 26 2020 11:59PM. To enhance the reproducibility of your results, we recommend that if applicable you deposit your laboratory protocols in protocols.io, where a protocol can be assigned its own identifier (DOI) such that it can be cited independently in the future. For instructions see: http://journals.plos.org/plosone/s/submission-guidelines#loc-laboratory-protocols

We look forward to receiving your revised manuscript.

Kind regards,

Anderson de Souza Sant'Ana, PhD

Academic Editor

PLOS ONE

Journal Requirements:

2) Please provide additional details regarding participant consent. In the ethics statement in the Methods and online submission information, please ensure that you have specified (1) whether consent was suitably informed and (2) what type you obtained (for instance, written or implied through submission of the questionnaire). If the need for consent was waived by the ethics committee, please include this information.

3) Please include additional information regarding the survey or questionnaire used in the study and ensure that you have provided sufficient details that others could replicate the analyses. If you developed and/or translated a questionnaire as part of this study and it is not under a copyright license more restrictive than Creative Commons Attribution (CC-BY), please include a copy, in both the original language and English, as Supporting Information.

4)  Please include in your Methods section the date ranges over which you recruited participants to this study, and please describe in more detail how recruitment was targeted.

5) Thank you for stating the following in the Acknowledgments Section of your manuscript:

[This work was supported by the Coordination for the Improvement of Higher Education Personnel (CAPES), the National Council for Scientific and Technological Development (CNPq), the Fundação de Amparo à Pesquisa do Estado de Minas Gerais (FAPEMIG) and the Pró-Reitoria de Pesquisa da UFMG (PRPq/UFMG).]

 [This worked was supported by the Minas Gerais Research Support Foundation

(Fundação de Amparo à Pesquisa do Estado de Minas Gerais - FAPEMIG), the

Brazilian Federal Agency for the Support and Evaluation of Graduate Education

(Coordenação de Aperfeiçoamento de Pessoal de Ensino Superior - CAPES), the

National Council for Scientific and Technological Development (Conselho Nacional de

Desenvolvimento Científico e Tecnológico - CNPq)]

Please include the updated Funding Statement in your cover letter. We will change the online submission form on your behalf.

Reviewers' comments:

Reviewer's Responses to Questions

**Comments to the Author**

1. Is the manuscript technically sound, and do the data support the conclusions?

Reviewer #1: Yes

Reviewer #2: Yes

Reviewer #3: Yes

2. Has the statistical analysis been performed appropriately and rigorously? 

Reviewer #1: Yes

Reviewer #2: Yes

Reviewer #3: Yes

3. Have the authors made all data underlying the findings in their manuscript fully available?

Reviewer #1: Yes

Reviewer #2: Yes

Reviewer #3: Yes

4. Is the manuscript presented in an intelligible fashion and written in standard English?

Reviewer #1: Yes

Reviewer #2: Yes

Reviewer #3: Yes

5. Review Comments to the Author

Reviewer #1: Relevance of Salmonella in foods should be adressed ( Food Control, Volume 104, October 2019, Pages 308-312; Int J dairy technology, Volume73, Issue1, February 2020, Pages 296-300; Journal of Dairy ScienceVolume 102, Issue 8August 2019Pages 6756-676).

The author should add more pratical consideration about the study,

The English is adaequate too.

Please decrease the introduction also.

Reviewer #2: The manuscript provides actual information regarding the use of processed feed against raw food in pets and the potential risk of zoonotic bacterial transmission and spread. Besides the useful information about how the owners understand the risks associated with raw meat and they care about this issue, the manuscript provides novel information about phenotypic and genetic characteristics of Salmonella, C. perfringens and C. difficile in feed and pets. According to the vision of this reviewer, the manuscript adds a valuable and useful information to the specific literature.

Writing style should be carefully reviewed. Please see that some time you use Clostridium difficile and others Clostridioides difficile. Also, there are several empty spaces before punctuations introduced during typing that should be corrected.

Reviewer #3: The authors present results mainly about important enteropathogens such as Salmonella spp., C. perfringens, and C. difficile, in feces of dogs fed different diets, raw meat-based diets (RMBD) or commercial dry feed. The topic is interesting. The abstract and the introduction are good and show all information about the work. The presented data support the conclusions

My observations are the next:

check the spaces

Line 110- specify times and conditions

6. PLOS authors have the option to publish the peer review history of their article (what does this mean?). If published, this will include your full peer review and any attached files.

Reviewer #1: No

Reviewer #2: Yes: Mariano Fernandez-Miyakawa

Reviewer #3: No

---

## [Author Response · Author response to Decision Letter 0]

18 Mar 2020

Editor

• Thank you for stating the following in the Acknowledgments Section of your manuscript “This work was supported by the Coordination for the Improvement of Higher Education Personnel (CAPES), the National Council for Scientific and Technological Development (CNPq), the Fundação de Amparo à Pesquisa do Estado de Minas Gerais (FAPEMIG) and the Pró-Reitoria de Pesquisa da UFMG (PRPq/UFMG). We note that you have provided funding information that is not currently declared in your Funding Statement. However, funding information should not appear in the Acknowledgments section or other areas of your manuscript. We will only publish funding information present in the Funding Statement section of the online submission form. Please remove any funding-related text from the manuscript and let us know how you would like to update your Funding Statement. Currently, your Funding Statement reads as follows: This worked was supported by the Minas Gerais Research Support Foundation (Fundação de Amparo à Pesquisa do Estado de Minas Gerais - FAPEMIG), the Brazilian Federal Agency for the Support and Evaluation of Graduate Education (Coordenação de Aperfeiçoamento de Pessoal de Ensino Superior - CAPES), the National Council for Scientific and Technological Development (Conselho Nacional de Desenvolvimento Científico e Tecnológico - CNPq). Please include the updated Funding Statement in your cover letter. We will change the online submission form on your behalf.

Authors: Corrected as required

• Please provide additional details regarding participant consent. In the ethics statement in the Methods and online submission information, please ensure that you have specified (1) whether consent was suitably informed and (2) what type you obtained (for instance, written or implied through submission of the questionnaire). If the need for consent was waived by the ethics committee, please include this information.

Authors: The participant consent was informed online just before the survey. This information can now be found in the Material and Methods section as suggested. 

• Please include additional information regarding the survey or questionnaire used in the study and ensure that you have provided sufficient details that others could replicate the analyses. If you developed and/or translated a questionnaire as part of this study and it is not under a copyright license more restrictive than Creative Commons Attribution (CC-BY), please include a copy, in both the original language and English, as Supporting Information.

Authors: The description of the survey was revised as suggested. Also, a copy of the questionnaire was added as a Supporting Information, so others can now replicate the analyses.

• Please include in your Methods section the date ranges over which you recruited participants to this study, and please describe in more detail how recruitment was targeted.

Authors: The requested date ranges were added, and the recruitment was added in more detail.

Reviewer 1

• Relevance of Salmonella in foods should be addressed (Food Control, Volume 104, October 2019, Pages 308-312; Int J dairy technology, Volume73, Issue1, February 2020, Pages 296-300; Journal of Dairy ScienceVolume 102, Issue 8August 2019Pages 6756-676).

Authors: Thank you for this suggestion. The text was revised and the relevance of Salmonella as a foodborne disease is now cited in text together with some new references.

• The author should add more practical consideration about the study.

Authors: Thanks for this suggestion. A more practical statement was added in the final paragraph of the manuscript.

• The English is adequate too.

Authors: Thank you for this comment

• Please decrease the introduction also.

Authors: Although we want to be concise throughout the paper, we were unable to reduce the introduction without compromising the quality of the manuscript.

Reviewer 2

• The manuscript provides actual information regarding the use of processed feed against raw food in pets and the potential risk of zoonotic bacterial transmission and spread. Besides the useful information about how the owners understand the risks associated with raw meat and they care about this issue, the manuscript provides novel information about phenotypic and genetic characteristics of Salmonella, C. perfringens and C. difficile in feed and pets. According to the vision of this reviewer, the manuscript adds a valuable and useful information to the specific literature

Authors: Thank you for this comment.

• Writing style should be carefully reviewed. Please see that some time you use Clostridium difficile and others Clostridioides difficile. Also, there are several empty spaces before punctuations introduced during typing that should be corrected.

Authors: All the manuscript was carefully revised to improve the writing style, grammar and mistyping. All changes are marked to make the revision easier.

Reviewer 3

• The authors present results mainly about important enteropathogens such as Salmonella spp., C. perfringens, and C. difficile, in feces of dogs fed different diets, raw meat-based diets (RMBD) or commercial dry feed. The topic is interesting. The abstract and the introduction are good and show all information about the work. The presented data support the conclusions.

Authors: Thank you for this comment.

• Check the spaces. 

Authors: All the manuscript was carefully revised to improve the writing style, grammar and mistyping.

---

## [Editor Report · Decision Letter 1]

20 Mar 2020

Fecal shedding of Salmonella spp., Clostridium perfringens, and Clostridioides difficile in dogs fed raw meat-based diets in Brazil and their owners’ motivation

PONE-D-20-01523R1

Dear Dr. Silva,

We are pleased to inform you that your manuscript has been judged scientifically suitable for publication and will be formally accepted for publication once it complies with all outstanding technical requirements.

With kind regards,

Anderson de Souza Sant'Ana, PhD

Academic Editor

PLOS ONE
---

## [Editor Report · Acceptance letter]

24 Mar 2020

PONE-D-20-01523R1 

Fecal shedding of *Salmonella* spp., *Clostridium perfringens*, and *Clostridioides difficile* in dogs fed raw meat-based diets in Brazil and their owners’ motivation 

Dear Dr. Silva:

I am pleased to inform you that your manuscript has been deemed suitable for publication in PLOS ONE. Congratulations! Your manuscript is now with our production department. 

With kind regards,

on behalf of

Professor Anderson de Souza Sant'Ana 

Academic Editor

PLOS ONE